# Low-Level Tetracycline Resistance Gene *tet(O)*_3 in *Campylobacter jejuni*

**DOI:** 10.3390/antibiotics12030426

**Published:** 2023-02-21

**Authors:** Cátia Pacífico, Marc M. S. M. Wösten, Friederike Hilbert

**Affiliations:** 1Institute of Food Safety, University of Veterinary Medicine, 1210 Vienna, Austria; 2Department of Biomolecular Health Sciences, Utrecht University, 3584 CL Utrecht, The Netherlands

**Keywords:** tetracycline resistance, epidemiological cutoff, foodborne pathogen, β-lactamase, multidrug efflux pump

## Abstract

*Campylobacter* (*C*.) spp. are the most important foodborne, bacterial, and zoonotic pathogens worldwide. Resistance monitoring of foodborne bacterial pathogens is an important tool to control antimicrobial resistance as a part of the “One Health” approach. The detection and functionality of new resistance genes are of paramount importance in applying more effective screening methods based on whole genome sequencing (WGS). Most tetracycline-resistant *C.* spp. isolates harbor *tet(O)*, a gene that encodes a ribosomal protection protein. Here we describe *tet(O)*_3, which has been identified in two food isolates of *C. jejuni* and is very similar to the *tet(O)* gene in *Streptococcus pneumoniae*, having a truncated promoter sequence. This gene confers resistance to tetracycline below 1 mg/L, which is the epidemiological cut-off value. We have analyzed the entire genome of these two isolates, together with a *C. jejuni* isolate found to have high-level resistance to tetracycline. In contrast to the highly resistant isolate, the promoter of *tet(O)*_3 is highly responsive to tetracycline, as observed by reverse transcription polymerase chain reaction (RT-PCR). In addition, the two isolates possess a CRISPR repeat, fluoroquinolone resistance due to the *gyrA* point mutation C257T, a β-lactamase resistance gene *bla_OXA-184_*, a multidrug efflux pump CmeABC and its repressor CmeR, but no plasmid. Low-level antibiotic resistant *C. jejuni* might therefore have an advantage for surviving in non-host environments.

## 1. Introduction

*Campylobacter* spp. are important zoonotic bacterial pathogens that are transmitted to humans through food, mainly contaminated poultry products and beef [1,2,3], and direct contact with animals, including production and companion animals, wildlife and water [4,5,6]. The consumption of improperly handled contaminated poultry products is also considered to be a major source of human infection [1]. Human campylobacteriosis is in most cases a self-limiting disease but can lead to chronic and autoimmune diseases such as reactive arthritis, inflammatory bowel disease, Guillain-Barre Syndrome, and other related diseases. Infections are mainly sporadic, but *Campylobacter* outbreaks are increasing worldwide [7,8]. Treatment with antibiotics is not usually indicated, but septicaemia or non-self-limiting diseases may require antibiotic therapy. Antimicrobial resistance (AMR) can therefore be a problem when devising treatment strategies. Resistance to macrolides and fluoroquinolones, the drugs recommended for treatment [9], is increasing in food and animal isolates [10]. Thus, monitoring AMR levels is as important in the “One Health” approach as identifying specific resistant clones that can spread between animals, humans and the environment [10].

AMR is a major and growing health problem worldwide. AMR testing is based on phenotypic methods using classical microbiology, such as agar and broth dilution assays that are standardized to define the appropriate antimicrobial therapy to combat the disease, using clinical breakpoints for human and animal pathogens [11,12]. However, the resistance monitoring of foodborne pathogens based on phenotypic testing is rather time-consuming. AMR surveillance data is expected to provide information on the amount and time course of antibiotic resistance in food, animal, human, and, more recently, environmental bacterial isolates promptly.

Therefore, AMR surveillance testing could be either replaced or complemented by genotypic testing, such as WGS of bacterial isolates or shotgun metagenomics of environmental, faecal or food samples to identify resistance genes and the general microbiome, rather than isolating different bacterial species [13,14]. Advantages are obvious, since by performing (i) WGS analysis of a bacterial isolate, not only can the antimicrobial resistance be detected, but also complete typing, the definition of virulence and metabolic genes, the mobile genetic elements and the tracking of human outbreak isolates can be achieved. Moreover, shotgun metagenomics can provide data on the entire microbiome and high-depth sequencing can be used to detect antimicrobial resistance genes, virulence genes and metabolic genes of all of the different microbial species in an individual sample. However, limitations should not be overlooked, as only previously known (or very similar) resistance genes can be detected, and precise, minimal, inhibitory concentrations are not defined for every resistance gene/bacteria combination.

The minimum inhibitory concentration (MIC—the in vitro defined minimum concentration of an antibiotic substance that is able to inhibit growth of a certain bacterium) is the basis for defining clinical and epidemiological breakpoints. The epidemiological cut-off value, defined by EUCAST, separates microorganisms with and without an acquired mechanism of resistance to a specific antimicrobial substance based on phenotypic testing [15]. With this method, mutated genes or genes that confer low-level resistance can be overlooked, as low-level resistance may be an important survival mechanism for *Campylobacter* in non-host environments such as water, feed or soil where residues or metabolites of low antibiotic activity might be present [16]. 

*C. jejuni* isolates are screened primarily for resistance to ciprofloxacin/nalidixic acid, erythromycin, gentamicin, tetracycline, and streptomycin. In addition, human isolates are optionally tested for resistance to amoxicillin/clavulanic acid, azithromycin, ertapenem, imipenem and meropenem [10]. High resistance rates are common for fluoroquinolones/quinolones, caused by a mutation in *gyrA* [10], and for tetracycline, based on *tet(O)*, which codes for a ribosomal protection protein [17]. Isolates that are phenotypically sensitive to tetracycline (based on the epidemiological cut-off value) and positive for *tet(O)* through sequence analysis have already been described [18].

Here, we describe two chicken meat isolates of *C. jejuni* that are phenotypically sensitive to tetracycline, at an epidemiological cut-off value of 1 mg/L, in which we could identify a novel *tet(O)* gene with a highly tetracycline-sensitive promoter. Their entire genome was analysed and made publicly available.

## 2. Results

Food isolates of *C. jejuni* from chicken meat were detected in a survey of foodborne pathogens from various types of meat in Austria. By testing a set of tetracycline susceptible isolates using genetic methods for the most common resistance genes by PCR, we identified that these two *C. jejuni* isolates were genotypically positive by *tet(O)*-PCR. These isolates were again screened for antimicrobial resistance using phenotypic testing methods and epidemiological and/or clinical breakpoints. Both isolates were tested for tetracycline susceptibility, using both microbroth dilution and disc diffusion assays, and showed a breakpoint of 1 mg/L and a zone inhibition of 36 and 34 mm, respectively. Further identification of this PCR product was done by PCR and Sanger sequencing of the adjacent genetic region. These isolates lack the typical upper *tet(O)* promoter region. For detailed analysis, both isolates (FC77 and FC88) and a highly tetracycline-resistant isolate of *C. jejuni* (GC119) were examined by WGS using Illumina technology. The total number of reads ranged from 1.46 M to 1.94 M per sample, yielding 97.0–97.3% high-quality reads. After assembly with SPAdes and removal of low-quality contigs, the assembled genomes had between 29 and 36 contigs, corresponding to a genome sequence length between 1.73 and 1.75 Mbp (Table 1). All isolates have between 1791 and 1814 coding sequences (CDS), 40 tRNAs, two rRNAs and one tmRNA. Interestingly, FC77 and FC88 have a CRISPR repeat in their sequences. 

The in silico identification of virulence and antibiotic resistance genes was performed with ABRicate, a bioinformatic tool for the bulk screening of contigs based on seven different AMR databases (Table 2). The *tet(O)* gene was identified in all three isolates by Resfinder, CARD, ARG-ANNOT and the NCBI database. Resfinder detected a *tet(O)*_3 in isolates FC77 and FC88 but a *tet(O)*_1 in GC119. 

A β-lactamase gene was identified in all three isolates by Resfinder, CARD and NCBI. In GC119 it was identified as *bla_OXA-61_*, but in FC77 and FC88 a *bla_OXA-184_* was detected. All three isolates were phenotypically identified as having low-level ampicillin resistance, with microdilution and disc-susceptibility testing yielding a MIC for GC119, FC77 and FC88 of 16 mg/L. 

The CARD database detected three membrane components of the multidrug efflux pump, CmeABC, and its repressor gene, *cmeR*, in all three isolates. 

A single point mutation, C257T, in the *gyrA* gene within the QRDR was also detected in all three isolates, resulting in an amino acid substitution in the gyrase A subunit at position 86, from threonine to isoleucine, which is known to result in phenotypic quinolone resistance. Consistent with this mutation, all three isolates showed phenotypic resistance to ciprofloxacin and nalidixic acid.

VFDB predicted that between 116 and 127 genes in the three isolates would be primarily associated with flagellar components (*fliADEFGHIKLMNPQRSWY*, *flgABCDEFGHIJKMPQRSW*, *ptmAB*, *flaABCDG*, *flhABGF*, *flgABCDEIJKMS*, *motAB*, *pflA*), lipid A biosynthesis (*eptC*, *htrB*), motility (*pseD/maf2*, *pseE/maf5*, *maf4*), toxin-coregulated pilus biosynthesis (*tcpI*), invasion (*ciaBC*), adhesion (*pebA*), lipoproteins (*jlpA*), capsule components (*kpsCDEFMST*, *cysC*, *Cj1416c*, *Cj1417c*, *Cj1419c*, *Cj1420c*, *Cj1421c*, *hddAC*, *gmhA2*, *fcI*, *rfbC*), membrane proteins (*cadF*, *porA*), chemotaxis (*cheA*, *cheV*, *cheV3*, *cheY*, *tlpA*, *cheW*), metabolism *(pseABCFGHI*, *gmhA*, *gmhB*, *hldD*, *hldE*, *neuA1*, *neuC1*, *neuB1*, *wlaN*, *Cj1135*, *Cj1136*, *Cj1138*, *Cj1137c*, *cstIII*), lipopolysaccharide (*waaC*, *waaF*, *waaV*), sigma factors (*rpoN*), toxins (*cdtABC*). For a complete list, see Appendix A.

When inspecting the sequences for predicted phages or prophages using PHASTER, two putative prophages were identified with high fidelity in the genome of the isolate GC119, while two low-scoring prophages were predicted for FC88, and one medium-scoring prophage was predicted for FC77.

Transcription of the *tet(O)* gene was detected in all three isolates by RT-PCR. The *tet(O)*_1 gene of isolate GC119 is 99.53% similar to the *tet(O)*_1 described for *C. jejuni* (GenBank: M18896.2). Experimental data also confirmed a tetracycline MIC of 64 mg/L and an “intact” promoter for *tet(O)*. Interestingly, the FC77 and FC88 *tet(O)*_3 gene is 99.64% similar to a *tet(O)* gene in *Streptococcus pneumoniae* (GenBank: Y07780.1), which contains a truncated form of the promoter region (Figure 1).

To analyze whether *tet(O)*_3 with the truncated promoter sequence is transcribed in FC77 and FC88, RNA was isolated from these isolates and the positive control isolate GC119. Bacterial cultures were harvested in their logarithmic growth phase, both without tetracycline and with the lowest level of tetracycline, 0.6 mg/L defined for isolates FC77 and FC88 with no growth reduction. Although GC119 was highly resistant to tetracycline, the same low concentration was used for the assay. As seen in Figure 2, the transcription of *tet(O)* was identified in all three isolates. In GC119, the highly resistant isolate harboring *tet(O)*_1, no effect of low-level concentration tetracycline on transcription was detected. In both FC77 and FC88 isolates, a significant transcriptional activation by tetracycline was detected by RT-PCR, with *tet(O)*_3 with the truncated promoter being activated by tetracycline at low concentrations (Figure 2). 

The sequence of the promoter region was analyzed in detail in comparison to the GC119 and database sequence from *C. jejuni* 81–176. The start codon was identified in all four isolates. The ribosomal binding site differed in sequence between 81–176 and GC119 by a base change from “AG” to “GA” and FC77, FC88 to “GG”. The sequence of the mRNA transcription site and the −10 region was consistent in all four sequences. Eight bp upstream of the −10 region, the sequence of FC77 and FC88 with GC119 and 81–176 began to diverge. By taking the spacing between the −10 and −35 region into account [19], the −35 regions of the *tet(O)* promoters of GC119 and 81–176 differ from those of FC77 and FC88 by four nucleotides (Figure 3). 

The protein sequences of the TetO protein of FC77, FC88, GC119, *C. jejuni* sequence M18896.2 and *S. pneumoniae* sequence Y07780.1 have been aligned. The protein sequences of FC77 and FC88 are 100% identical and show 99.53% identity to the *S. pneumoniae* protein and 98.12% identity to the *C. jejuni* protein sequence, respectively (Appendix A). 

## 3. Discussion

*C. jejuni* is one of the most important foodborne pathogens. As such, antimicrobial resistance in this pathogen is not only an important cause of possible treatment failure but also a direct link of transmission between food-producing animals and humans. Therefore, screening and surveillance of antimicrobial resistance of *C. jejuni* in human, animal, and food isolates are mandatory in the European Union [8]. Resistance to quinolones is most often reported, mainly based on point mutations, while tetracycline resistance is the second most commonly reported, based on acquired genes located on plasmids or in the nucleoid [20]. In addition to a possible increased need for tetracycline as a treatment option, environmental screenings confirm that tetracycline and its metabolites are widespread, as well as tetracycline-resistant bacteria [21]. As one of the first resistance mechanisms studied in detail, tetracycline resistance genes now offer the opportunity to study evolutionary aspects in bacteria based on sequence data from the entire genome [22].

Tetracycline resistance in bacteria is based on three different mechanisms. Drug efflux pumps Tet (A-L, V, Y) have been described mainly in gram-negative bacterial species. These genes are often directly regulated by a repressor gene, *tetR*, that responds to tetracycline. This response regulator has been used extensively for studies on artificial gene regulation. Resistance mechanisms due to the enzymatic inactivation of tetracycline have also been described in *Escherichia coli* with high-level tigecycline resistance, a third-generation tetracycline. Ribosomal protection proteins are widespread in environmental bacteria and Gram-positive bacteria. They can weaken the binding of tetracycline to the ribosome and release the drug [23,24]. Tetracycline resistance genes identified in *C. jejuni* so far were all based on a ribosomal binding protein *tet(O)*, and a mosaic gene *tet(O/32/O)* recently described [18,25]. Several authors stated that the phenotypic and genotypic detection of these resistance genes is well correlated, with many of them reporting a 100% correlation [26,27,28,29,30]. EUCAST gives an epidemiological breakpoint for tetracycline of ≤1 mg/L in *C. jejuni*. Therefore, isolates with resistance to tetracycline at or above 2 mg/L are considered resistant and harbor a resistance determinant, and isolates resistant to tetracycline at or below 1 mg/L are considered wild-type and should not carry a resistance object. Nonetheless, Webb and co-workers defined an epidemiological cut-off value of 0.5 mg/L for tetracycline using a tool from Turnidge et al. [31,32]. Moreover, the genes that Dahl and co-workers described as non-functional for *tet(O)* in *Campylobacter*, due to a frameshift mutation, can be silenced [18]. These frameshift mutations have only been identified in sequence types (ST) 21, 50 and 8873. Here we describe a functional *tet(O)* gene in two *Campylobacter* food isolates both from ST 3015, that confers resistance to tetracycline at the epidemiological break point of 1 mg/L. This gene has a higher homology to a *tet(O)* in *Streptococcus pneumoniae* (GenBank: Y07780.1.) than to the *tet(O)* gene in the *C. jejuni* reference genome (GenBank: M18896.2). The designation with Resfinder is *tet(O)*_3 versus *tet(O)*_1 of *C. jejuni*. This gene responds very well to low levels of tetracycline (0.6 mg/L), whereas in a control isolate GC119, the *tet(O)*_1 gene homologous to the *C. jejuni* reference genome does not respond to such low levels of tetracycline.

Detailed sequence analysis revealed major differences starting 9bp upstream of the −10 region, which corresponds to a change in the promoter region that could explain both the low tetracycline resistance and response to low tetracycline levels (Figure 3). The genome position of *tet(O)*_3 in *C. jejuni* FC77 and FC88 is chromosomal, as is the *tet(O)*_1 gene of our control isolate GC119. However, the location within the genome differs from GC116. The *tet(O)*_3 is located between two truncated genes; *feoA* and *feoB*. Both functional genes are necessary for iron uptake and this could be indicative of horizontal gene transfer events since both isolates harbor a functional extra copy of these two genes. The horizontal gene transfer of tetracycline resistance in *C.* spp. has been described and confirmed in vitro [33].

In addition, based on Resfinder and CARD databases, all three isolates, both FC77 and FC88 and the highly resistant control isolate GC119, contained a *bla_OXA_* gene encoding a β-lactamase; while in FC77 and FC88 this gene was identified as *bla_OXA-184_*, however, it was *bla_OXA-61_* in GC119. Both of these genes have been described in *C. jejuni* and confer ampicillin resistance [34,35].

The CmeABC efflux pump and its CmeR repressor (a repressor of the *tetR*-like family) were identified by CARD and Resfinder in all three isolates. Overexpression of this efflux pump and mutations in its repressor have been discussed for high tetracycline resistance levels and the selection of quinolone resistance [33,36]. An effect of reducing tetracycline resistance or possible involvement in regulation by tetracycline has not been reported so far.

In general, foodborne pathogens must withstand different environmental conditions, such as the intestinal tract of the animal and human host, where they may be exposed to high concentrations of antimicrobial agents, or the food storage, which exposes them to cold shock due to refrigeration, or the wastewater that brings them together with various competing microorganisms. *Campylobacter* spp. have a rather small genome and need to adapt to these environments [16], often through horizontal gene transfer. Here we speculate that a low tetracycline resistance, which responds strongly to low tetracycline levels, might confer an evolutionary advantage in environments contaminated with low levels of this antibiotic substance, such as effluent water, sewage treatment plants, or surface water. As an example, exposure to antibiotics well below the epidemiological or clinical breakpoint can serve to spread and lead to the persistence of resistance genes [37,38].

## 4. Materials and Methods

The isolation of *Campylobacter* spp. from chicken meat was performed using ISO 10272-1:2017 with minor modifications. A meat sample of 25 g was inoculated in Bolton broth (Oxoid CM983 with supplement SR208E, Basingstoke, England) for 48 h at 42 °C under microaerobic conditions (Oxoid Gas Generating Kit Campylobacter System BR060A, Basingstoke, England). The enrichment was streaked onto modified CCDA agar (Oxoid CM739 and supplement SR155E) and CampyFoodAgar (bioMerieux, Marcy l’Etoile, France) and incubated at 42 °C under microaerobic conditions for 48 h. Colonies selected based on colony morphology were verified for *C. jejuni* or *C. coli* species using PCR detection [39].

Antimicrobial resistance testing was performed using the standard CLSI M45 method for dilution and disk susceptibility testing. For the disk diffusion susceptibility testing, MH-plates (Oxoid, CM337) with 5% lyophilized horse blood (SR050B, Oxoid) were used with standard discs (Oxoid Antimicrobial Susceptibility Test Discs) applied with a disc dispenser. The dilution test was performed in MH broth (CM0405, Oxoid) with 5% lyophilized horse blood using tetracycline (0.002–512 mg/L), ampicillin (0.03–512 mg/L), and ciprofloxacin (0.002–256 mg/L) incubated for 24–48 h at 42 °C under microaerobic conditions. Clinical breakpoints for *C. jejuni* isolates were taken from CLSI M45 guidelines [9], and EUCAST epidemiological breakpoints [12,15].

Genomic DNA was extracted from the colonies grown on MH-Plates with 5% lyophilized horse blood for 48 h at 42 °C under microaerobic conditions. DNA was extracted using a QIAamp DNA mini kit (Qiagen, Venlo, The Netherlands). RNA extraction was performed on *Campylobacter* cells grown in MH broth with 5% lyophilized horse blood with and without 0.6 mg/L tetracycline to the mid-log growth phase at OD600 0.6. RNA extraction was performed according to Peqlab protocol with Peq Gold Tri Fast (VWR, 30–2010, Randor, PA, USA) and DNase treatment using 5U DNase I (PerfeCta, Quanta Biosciences 95150-01K, Beverly, MA, USA). Reverse transcription was performed using Tetro c DNA synthesis Lit (Bioline, BIO-65042, Cincinnati, OH, USA) and immediately prepared for RT-PCR using SensiFAST^TM^ SYBR No-ROX Kit (Bioline, BIO-98002) on Magnetic Induction Cycler (68MIC-2, BMS, Hessisch Oldendorf, Germany) used with micPCR software version 2.10.3. For *tet(O)* primers tetO-for 5′-AACTTAGGCATTCTGGCTCAC-3′ and tetO-rev 5′-TCCCACTGTTCCATATCGTCA-3′ were used. As a control, 16S RNA RT-PCR with primers com-for 5′-CAGCAGCCGCGGTAATAC-3′ com-rev 5′-CCGTCAATTCCTTTGAGTTT-3′ was used.

WGS library preparation (Nextera XT library prep lit Illumina, San Diego, CA, USA) was done using 2 ng of DNA. DNA sequencing was performed on an Illumina HiSeq 2500 platform using a 2 × 250 bp paired-end approach and the trimming of the reads was done at MicrobesNG (Birmingham, UK). The trimmed reads were analyzed by the INNUca pipeline [40] and read quality was checked using FastQC (https://www.bioinformatics.babraham.ac.uk/projects/fastqc/, accessed on 31 January 2023) and trimmomatic version 0.38 [41]. De novo assembly was performed after paired reads were merged with PEAR [42] version 0.9.10 using SPAdes [43] version 3.13 using the “careful” option. Output contigs with less than one-third of the assembly mean coverage or <10× were removed and assembly was corrected by Pilon [44] version 1.18. Detection of antimicrobial resistance and virulence genes was performed using ABRicate (v 0.8) bundled with CARD, Resfinder, ARG-ANNOT, NCBI, and VFDB databases. Phage and prophage sequences were identified using Phaster (https://phaster.ca, accessed on 31 January 2023). MAUVE Contig Mover (v 2.4.0) was used to orientate and order contigs using as reference genome *C. jejuni* subsp. *jejuni* strain MTVDSCj07 (NCBI accession number CP017031.1) or *C. jejuni* subsp. *jejuni* strain MTVDSCj16 (NCBI accession number CP017033.1).

Raw sequence data were deposited in Sequence Read Archive (SRA) from NCBI with the accession number PRJNA558489.

## 5. Conclusions

Alternative methods such as shotgun metagenomics or targeted sequencing for antimicrobial resistance tests, especially for screening and monitoring, are currently under discussion. Current phenotypic testing cannot monitor resistance determinants that express a low level of resistance below current breakpoints (clinical and epidemiological). Such genes could be important for horizontal transfer under low concentrations of antimicrobial agents that can be found in environments such as sewage and surface water, sewage treatment plants or soil. Although we used the current epidemiological cut-off value for *C. jejuni*, we still identified two food isolates that are sensitive to tetracycline and have a functional gene encoding resistance to low levels of tetracycline, which is highly responsive to the antibiotic. Therefore, we envision that sequence-based methods for antimicrobial resistance surveillance and screening will be deeply relevant, particularly in non-host environments.

## Figures and Tables

**Figure 1 antibiotics-12-00426-f001:**
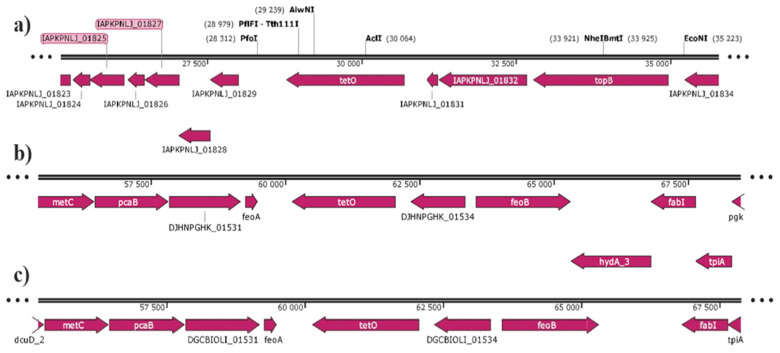
A genome analysis of the region around *tet(O)*_1 and *tet(O)*_3 of (**a**) GC119, (**b**) FC77; (**c**) FC88. In FC77 and FC88 the *tet(O)*_3 is located between two truncated virulence genes, *feoA* and *feoB.* Both isolates possess another copy of the non-truncated genes in their genome.

**Figure 2 antibiotics-12-00426-f002:**
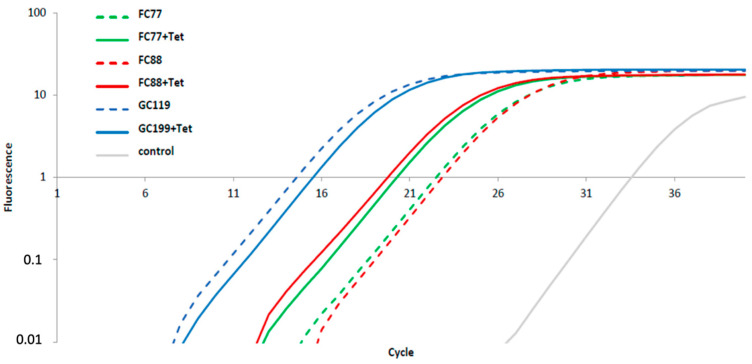
RT-PCR on *C. jejuni* FC77 (green), FC88 (red), and GC119 (blue) grown in MH broth plus 5% lyophilized horse blood with and without the addition of 0.6 mg/L tetracycline (both bolt and dashed line) harvested at mid logarithmic growth condition. As a control, a 16S RNA RT-PCR was included with the same sample extracts.

**Figure 3 antibiotics-12-00426-f003:**
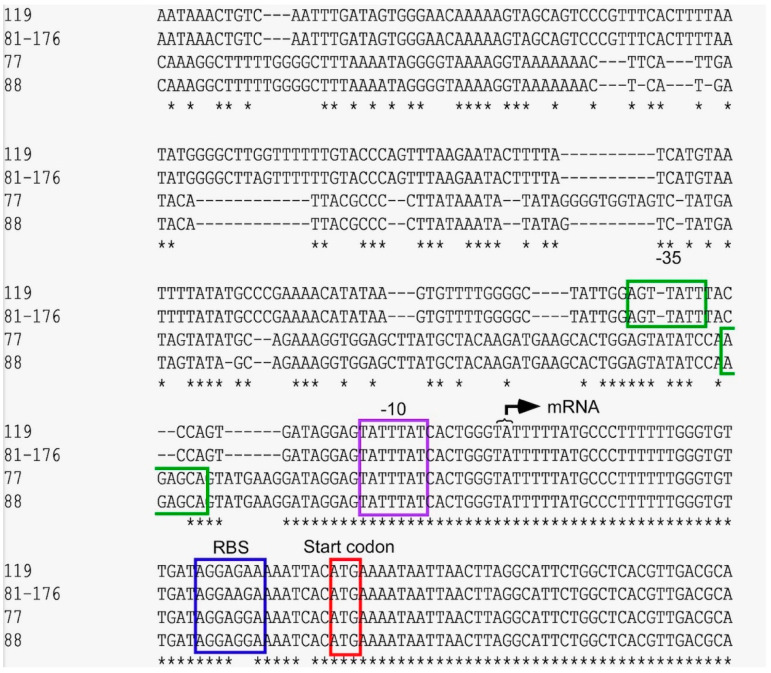
The promoter analysis of *tet(O)*_1 in GC119 and *Campylobacter* 81–176 and *tet(O)*_3 of FC77 and FC88. The “ATG” start codon of the protein is boxed in red, the ribosomal binding site in blue, the purple box indicates the −10 region, and the proposed −35 region is boxed in green; * shows sequence homology.

**Table 1 antibiotics-12-00426-t001:** The basic genome composition of the sequenced isolates.

*Campylobacter* Isolate	GC119	FC77	FC88
Origin	chicken meat	chicken meat	chicken meat
Number of contigs	29	34	36
Full length (bp)	1,754,293	1,733,738	1,733,513
GC content (%)	30	30	30
Coding sequences	1814	1791	1791
tRNA	40	40	40
rRNA	2	2	2
tmRNA	1	1	1
mRNA	1857	1834	1834
CRISPR repeats	0	1	1
MLST profile	760	3015	3015

**Table 2 antibiotics-12-00426-t002:** The virulence and antimicrobial resistance genes of the sequenced isolates.

	Resfinder	CARD	ARG-ANNOT	EcOH	NCBI	PlasmidFinder	VFDB *
GC119	*tet(O)*_1 *bla_OXA-61_*	*tet(O)* *bla_OXA-61_* *cmeABC* *cmeR*	*tet(O)* *bla_OXA-61_*	*-*	*tet(O)* *bla_OXA-61_*	*-*	116 genes
FC77	*tet(O)*_3 *bla_OXA-184_*	*tet(O)* *bla_OXA-184_* *cmeABC* *cmeR*	*tet(O)*	*-*	*tet(O)* *bla_OXA-184_*	*-*	125 genes
FC88	*tet(O)*_3 *bla_OXA-184_*	*tet(O)* *bla_OXA-184_* *cmeABC* *cmeR*	*tet(O)*	*-*	*tet(O)* *bla_OXA-184_*	*-*	127 genes

* Detailed results can be found in Appendix A.

## Data Availability

Raw sequence data were deposited in Sequence Read Archive (SRA) from NCBI with the accession number PRJNA558489.

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
