# Peer review of "Low-Level Tetracycline Resistance Gene tet(O)_3 in Campylobacter jejuni"

_antibiotics, 2023, doi:10.3390/antibiotics12030426_

Round 1
Reviewer 1 Report
The article 'Low level tetracycline resistance gene tet(O)_3 in Campylobacter jejuni ' provides a clear-cut analysis of the novel gene 'tet(O)', by analyzing its entire genome.
Very few corrections and suggestions were there.
- Title: Low level --> Low-level?
- Consider a graphical abstract / AMR mechanism, it will be easy to understand the multidrug binding and the efflux process.
Author Response
Dear reviewer,
Many thanks for your comments and suggestions. We have added all details below and revised the manuscript.
The article 'Low level tetracycline resistance gene tet(O)_3 in Campylobacter jejuni ' provides a clear-cut analysis of the novel gene 'tet(O)', by analyzing its entire genome.
Very few corrections and suggestions were there.
- Title: Low level --> Low-level?
Thanks for the suggestion. We changed it in the title and throughout the manuscript.
- Consider a graphical abstract / AMR mechanism, it will be easy to understand the multidrug binding and the efflux process.
We have added a graphical abstract on tet(O) resistance mechanism
Reviewer 2 Report
Pacífico et al. identified tet(O)_3 in the C. jejuni isolates FC77 and FC88. They also identified tet(O)_1 in the C. jejuni isolate GC119. They concluded that 1). tet(O)_3 conferred a low level of Tc resistance, and 2). tet(O)_1 conferred a high level of Tc resistance. They performed WGS on these three isolates. They concluded the lower Tc resistance conferred by tet(O)_3 compared to tet(O)_1 is because the former had a truncated promoter region compared to the latter.
Major comments:
1. Discussion regarding Tc needs to be furthered. Include the following: 1). discuss the Tc resistome (cite 10.1007/s00018-009-0172-6), 2). summarize works where TcR genes were identified, and 3). review evolutionary works where TcR genes were exposed to Tc pressures and led to various Tc resistance levels.
2. I am not convinced that the few nt changes (Figure 3) in the promoter region of tet(O)_1 and tet(O)_3 led to the difference in the Tc resistance level. To prove this, for each of tet(O)_1 and tet(O)_3, 1). synthesize the full Tc resistance cassette from de novo, 2). insert the cassette into the chromosome of a naïve Campylobacter strain, and 3). test the Tc resistance of the genetically engineered strain. To further elaborate on this experiment:
- The cassette should not be cloned from the FC77, FC88, and GC119 gDNA.
- Also try to insert the cassette into a commonly used E. coli chromosome.
Minor comments:
1. Line 21, "CRISP" → "CRISPR".
2. Line 24, "could Campylobacter spp. enable" → "could enable Campylobacter spp."
3. Line 29, remove "important". Try to avoid words with no actual meanings.
4. Lines 33–34, what does "characteristically" mean?
5. Line 48, what does "on the level" mean?
6. Line 63, define "MIC".
7. Line 66, "muted"?
8. Line 74, "-" → "—". Use em dash (—), not hyphen (-).
9. Line 85, "Both" → "Two".
10. Line 87, "tested tetracycline susceptibility" → "tested for tetracycline susceptibility".
11. Line 92, "sequencing and PCR" → "PCR and Sanger sequencing"?
12. Lines 93–94, remove "For detailed analysis".
13. I am not doing text editing any further. The manuscript needs extensive editing.
Author Response
Dear reviewer,
Thanks you for your suggestions and comments.
Comments and Suggestions for Authors
Pacífico et al. identified tet(O)_3 in the C. jejuni isolates FC77 and FC88. They also identified tet(O)_1 in the C. jejuni isolate GC119. They concluded that 1). tet(O)_3 conferred a low level of Tc resistance, and 2). tet(O)_1 conferred a high level of Tc resistance. They performed WGS on these three isolates. They concluded the lower Tc resistance conferred by tet(O)_3 compared to tet(O)_1 is because the former had a truncated promoter region compared to the latter.
Major comments:
- Discussion regarding Tc needs to be furthered. Include the following: 1). discuss the Tc resistome (cite 10.1007/s00018-009-0172-6), 2). summarize works where TcR genes were identified, and 3). review evolutionary works where TcR genes were exposed to Tc pressures and led to various Tc resistance levels.
Thanks for your suggestions we included a new discussion section regarding the Tc resistome and TcR genes and their response to tetracycline
- I am not convinced that the few nt changes (Figure 3) in the promoter region of tet(O)_1and tet(O)_3led to the difference in the Tc resistance level. To prove this, for each of tet(O)_1 and tet(O)_3,
We have shown in a reverse RT PCR that both tetO_1 and tetO_3 are transcribed. There is information that transcription is dependent on the promoter region in Campylobacter and many other bacteria and that already single point mutations can reduce, abolish, or elevate transcription of a gene (please refer to 10.1007/s00284-013-0420-8; 10.1128/JB.05493-11; 10.1128/JB.180.3.594-599.1998). We have adapted figure 3 as the the proposed position of the -35 region in the tetO of strains FC77 and FC88 was to far away from the -10 region. Spacing between the -10 and -35 region must be between 15-19 bp (Wösten, M.M.S.M. (1998). Eubacterial sigma factors. FEMS Microbiol. Rev. 22:127-150.). The -35 regions of the tetO promoters of GC119 and 81-176 differ from those of FC77 and FC88 by four nucleotides. We have added a short paragraph in the discussion section about the importance of the promoter sequence.
We cannot entirely role out that small changes in the TetO protein may be responsible for different levels of resistance, but based on the protein analysis it is not very likely, therefore we included in suppl. table 2 a comparison of the protein sequence of TetO_1, TetO-3 and the S. pneumoniae TetO protein and added an additional paragraph in the results section.
1). synthesize the full Tc resistance cassette from de novo, 2). insert the cassette into the chromosome of a naïve Campylobacter strain, and 3). test the Tc resistance of the genetically engineered strain. To further elaborate on this experiment:
- The cassette should not be cloned from the FC77, FC88, and GC119 gDNA.
- Also try to insert the cassette into a commonly used E. coli chromosome.
Suggested experiments have already been performed by other authors on the tetO_1 (please refer to 10.1016/j.febslet.2008.03.023 and 10.1016/0378-1119(88)90576-8). To get more insight into the gene’s function and/or transcription of tetO_3 we do not see an additional value by suggested experiments as 1. The gene has been found in two different isolates of C. jejuni from different meat samples with the same tetracycline resistance phenotype and transcription rate. 2. As E. coli and C. jejuni are different and use different kind of promoters (for information see 10.1128/JB.180.3.594-599.1998) we do not expect new insights in the gene’s function by inserting the gene into E. coli. Furthermore, the gene is 99.64% similar to the tetracycline resistance gene in Streptococcus pneumoniae (GenBank: Y07780.1) as stated in the manuscript, it is likely that the gene confers tetracycline resistance in other bacteria than C. jejuni as well.
Minor comments:
- Line 21, "CRISP" → "CRISPR". changed
- Line 24, "could Campylobacter spp. enable" → "could enable Campylobacter spp." changed
- Line 29, remove "important". Try to avoid words with no actual meanings. removed
- Lines 33–34, what does "characteristically" mean? Rephrased “in most cases”
- Line 48, what does "on the level" mean? We rephrased the sentence
- Line 63, define "MIC". defined
- Line 66, "muted"? changed to “mutated”
- Line 74, "-" → "—". Use em dash (—), not hyphen (-). changed
- Line 85, "Both" → "Two". changed
- Line 87, "tested tetracycline susceptibility" → "tested for tetracycline susceptibility". changed
- Line 92, "sequencing and PCR" → "PCR and Sanger sequencing"? changed
- Lines 93–94, remove "For detailed analysis". removed
- I am not doing text editing any further. The manuscript needs extensive editing. Text has been revised by a native English speaker.
Reviewer 3 Report
Dear Authors,
The manuscript entitled "Low level tetracycline resistance gene tet(O)_3 in Campylobacter jejuni" was reviewed.
The article is very important since it highlights a very attractive topic related to the bacterial resistance toward antibiotics. Mainly the resistance of Campylobacter jejuni against a low level of tetracycline. The article studies and analyses via molecular technics the detection of a new tet(O) gene in two food isolates of Campylobacter jejuni that confer to this bacterium a resistance against a low level of tetracycline, this gene is called tet(O)_3.
The article is well prepared and presented, the English language is good, different sections are well presented.
Kindly find below my comments and questions related to your work:
Minor (Comments and Questions):
01- For the whole manuscript, Authors are invited to use Campylobacter jejuni followed by (C. jejuni) in the first time they use the name of this species. Then they have to use C. jejuni instead of Campylobacter jejuni in all the sections of this article.
02- In the Abstract section, Line 19, authors are invited to add two between "these" and "isolates", the sentence will be as "these two isolates".
03- In the Abstract section, Lines 23-24, authors are invited to paraphrase the last sentence "Low-level antibiotic...". Since it is not very clear for readers.
04- In the Abstract section, Line 24, authors are invited to replace "Campylobacter spp." by "Campylobacter jejuni".
05- In the Keywords section, authors are invited to add Campylobacter jejuni to the list of keywords.
06- In the Introduction section, Line 30, "... poultry product and beef" authors are invited to add the following reference: (Prevalence, antimicrobial resistance and risk factors for campylobacteriosis in Lebanon).
07- In the Introduction section, Line 36, "... outbreaks are increasing" authors are invited to add the following reference: (Prevalence, laboratory findings and clinical characteristics of campylobacteriosis agents among hospitalized children with acute gastroenteritis in Lebanon).
08- In the Introduction section, Line 37, authors are invited to correct "septicemia".
09- In the Results section, Lines 116-120 and Lines 131-134, these two results are not shown in figures or tables in the manuscript authors are invited to add figures or tables related to these results.
10- In the Table 2, last column (VFDB), authors are requested to explain the meaning of the star after the word genes (116 genes *).
11- In the Discussion section, Line 217, authors are invited to correct the word "response".
Major (Comments and Questions):
01- Authors are invited to show the similarity percentage between the tet(O)_3 detected in the two isolates FC77 and FC88 with the tet(O)_1 described for C. jejuni in GenBank.
Best Regards,
Author Response
Dear reviewer,
Thanks for your valuable comments and suggestions. Please find the modified version of the manuscript uploaded and the answer to your comments below:
The manuscript entitled "Low level tetracycline resistance gene tet(O)_3 in Campylobacter jejuni" was reviewed.
The article is very important since it highlights a very attractive topic related to the bacterial resistance toward antibiotics. Mainly the resistance of Campylobacter jejuni against a low level of tetracycline. The article studies and analyses via molecular technics the detection of a new tet(O) gene in two food isolates of Campylobacter jejuni that confer to this bacterium a resistance against a low level of tetracycline, this gene is called tet(O)_3.
Thank you.
The article is well prepared and presented, the English language is good, different sections are well presented.
Thank you.
Kindly find below my comments and questions related to your work:
Many thanks for your comments
Minor (Comments and Questions):
01- For the whole manuscript, Authors are invited to use Campylobacter jejuni followed by (C. jejuni) in the first time they use the name of this species. Then they have to use C. jejuniinstead of Campylobacter jejuni in all the sections of this article.
Thanks for the suggestion. The manuscript has been adapted accordingly.
02- In the Abstract section, Line 19, authors are invited to add two between "these" and "isolates", the sentence will be as "these two isolates".
Thanks, Done.
03- In the Abstract section, Lines 23-24, authors are invited to paraphrase the last sentence "Low-level antibiotic...". Since it is not very clear for readers.
Sentence was rephrased.
04- In the Abstract section, Line 24, authors are invited to replace "Campylobacter spp." by "Campylobacter jejuni".
Changed.
05- In the Keywords section, authors are invited to add Campylobacter jejuni to the list of keywords.
We did add the name in the keywords as it is mentioned in the title already.
06- In the Introduction section, Line 30, "... poultry product and beef" authors are invited to add the following reference: (Prevalence, antimicrobial resistance and risk factors for campylobacteriosis in Lebanon).
Reference was added.
07- In the Introduction section, Line 36, "... outbreaks are increasing" authors are invited to add the following reference: (Prevalence, laboratory findings and clinical characteristics of campylobacteriosis agents among hospitalized children with acute gastroenteritis in Lebanon).
Reference was added.
08- In the Introduction section, Line 37, authors are invited to correct "septicemia".
English based on Latin spelling “septicaemia”
09- In the Results section, Lines 116-120 and Lines 131-134, these two results are not shown in figures or tables in the manuscript authors are invited to add figures or tables related to these results.
Results line 116-120 are shown in table 2; and those in line 131-134 in the suppl. data
10- In the Table 2, last column (VFDB), authors are requested to explain the meaning of the star after the word genes (116 genes *).
Added a foot note to the table
11- In the Discussion section, Line 217, authors are invited to correct the word "response".
Thanks for spotting. Changed.
Major (Comments and Questions):
01- Authors are invited to show the similarity percentage between the tet(O)_3 detected in the two isolates FC77 and FC88 with the tet(O)_1 described for C. jejuni in GenBank.
A comparison was done on the protein sequence of both proteins and added to Suppl. Table 2
Round 2
Reviewer 2 Report
Thank you for addressing my comments. I have no further questions.
Reviewer 3 Report
Dear Authors,
Your manuscript was re-reviewed,
I would like to thank you for the effort you made in the new version.
The article is better in its present form.
Thank you,
Best Regards,